# Bidirectional Recurrent Neural Networks as Generative Models

**Mathias Berglund**
Aalto University, Finland

**Tapani Raiko**
Aalto University, Finland

**Mikko Honkala**
Nokia Labs, Finland

**Leo Kärkkäinen**
Nokia Labs, Finland

**Akos Vetek**
Nokia Labs, Finland

**Juha Karhunen**
Aalto University, Finland

## Abstract

Bidirectional recurrent neural networks (RNN) are trained to predict both in the positive and negative time directions simultaneously. They have not been used commonly in unsupervised tasks, because a probabilistic interpretation of the model has been difficult. Recently, two different frameworks, GSN and NADE, provide a connection between reconstruction and probabilistic modeling, which makes the interpretation possible. As far as we know, neither GSN or NADE have been studied in the context of time series before. As an example of an unsupervised task, we study the problem of filling in gaps in high-dimensional time series with complex dynamics. Although unidirectional RNNs have recently been trained successfully to model such time series, inference in the negative time direction is non-trivial. We propose two probabilistic interpretations of bidirectional RNNs that can be used to reconstruct missing gaps efficiently. Our experiments on text data show that both proposed methods are much more accurate than unidirectional reconstructions, although a bit less accurate than a computationally complex bidirectional Bayesian inference on the unidirectional RNN. We also provide results on music data for which the Bayesian inference is computationally infeasible, demonstrating the scalability of the proposed methods.

## 1 Introduction

Recurrent neural networks (RNN) have recently been trained successfully for time series modeling, and have been used to achieve state-of-the-art results in supervised tasks including handwriting recognition [12] and speech recognition [13]. RNNs have also been used successfully in unsupervised learning of time series [26, 8].

Recently, RNNs have also been used to generate sequential data [1] in a machine translation context, which further emphasizes the unsupervised setting. Bahdanau *et al.* [1] used a bidirectional RNN to encode a phrase into a vector, but settled for a unidirectional RNN to decode it into a translated phrase, perhaps because bidirectional RNNs have not been studied much as generative models. Even more recently, Maas *et al.* [18] used a deep bidirectional RNN in speech recognition, generating text as output.

Missing value reconstruction is interesting in at least three different senses. Firstly, it can be used to cope with data that really has missing values. Secondly, reconstruction performance of artificially missing values can be used as a measure of performance in unsupervised learning [21]. Thirdly, reconstruction of artificially missing values can be used as a training criterion [9, 11, 27]. While traditional RNN training criterions correspond to one-step prediction, training to reconstruct longer gaps can push the model towards concentrating on longer-term predictions. Note that the one-step

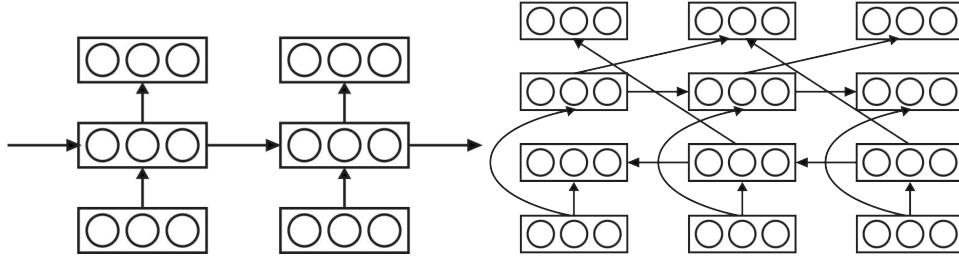

Figure 1: Structure of the simple RNN (left) and the bidirectional RNN (right).

prediction criterion is typically used even in approaches that otherwise concentrate on modelling long-term dependencies [see e.g. 19, 17].

When using unidirectional RNNs as generative models, it is straightforward to draw samples from the model in sequential order. However, inference is not trivial in smoothing tasks, where we want to evaluate probabilities for missing values in the middle of a time series. For discrete data, inference with gap sizes of one is feasible - however, inference with larger gap sizes becomes exponentially more expensive. Even sampling can be exponentially expensive with respect to the gap size.

One strategy used for training models that are used for filling in gaps is to explicitly train the model with missing data [see e.g. 9]. However, such a criterion has not to our knowledge yet been used and thoroughly evaluated compared with other inference strategies for RNNs.

In this paper, we compare different methods of using RNNs to infer missing values for binary time series data. We evaluate the performance of two generative models that rely on bidirectional RNNs, and compare them to inference using a unidirectional RNN. The proposed methods are very favourable in terms of scalability.

## 2 Recurrent Neural Networks

Recurrent neural networks [24, 14] can be seen as extensions of the standard feedforward multilayer perceptron networks, where the inputs and outputs are sequences instead of individual observations.

Let us denote the input to a recurrent neural network by $\mathbf{X} = \{\mathbf{x}_t\}$ where $\mathbf{x}_t \in \mathbb{R}^N$ is an input vector for each time step $t$. Let us further denote the output as $\mathbf{Y} = \{\mathbf{y}_t\}$ where $\mathbf{y}_t \in \mathbb{R}^M$ is an output vector for each time step $t$. Our goal is to model the distribution $P(\mathbf{Y}|\mathbf{X})$. Although RNNs map input sequences to output sequences, we can use them in an unsupervised manner by letting the RNN predict the next input. We can do so by setting $\mathbf{Y} = \{\mathbf{y}_t = \mathbf{x}_{t+1}\}$.

### 2.1 Unidirectional Recurrent Neural Networks

The structure of a basic RNN with one hidden layer is illustrated in Figure 1, where the output $\mathbf{y}_t$ is determined by

$$P\left(\mathbf{y}_t \mid \{\mathbf{x}_d\}_{d=1}^t\right) = \phi\left(\mathbf{W}_\mathrm{y}\mathbf{h}_t + \mathbf{b}_\mathrm{y}\right) \tag{1}$$

where

$$\mathbf{h}_t = \tanh\left(\mathbf{W}_\mathrm{h}\mathbf{h}_{t-1} + \mathbf{W}_\mathrm{x}\mathbf{x}_t + \mathbf{b}_\mathrm{h}\right) \tag{2}$$

and $\mathbf{W}_\mathrm{y}$, $\mathbf{W}_\mathrm{h}$, and $\mathbf{W}_\mathrm{x}$ are the weight matrices connecting the hidden to output layer, hidden to hidden layer, and input to hidden layer, respectively. $\mathbf{b}_\mathrm{y}$ and $\mathbf{b}_\mathrm{h}$ are the output and hidden layer bias vectors, respectively. Typical options for the final nonlinearity $\phi$ are the softmax function for classification or categorical prediction tasks, or independent Bernoulli variables with sigmoid functions for other binary prediction tasks. In this form, the RNN therefore evaluates the output $\mathbf{y}_t$ based on information propagated through the hidden layer that directly or indirectly depends on the observations $\{\mathbf{x}_d\}_{d=1}^t = \{\mathbf{x}_1, \ldots, \mathbf{x}_t\}$.

## 2.2 Bidirectional Recurrent Neural Networks

Bidirectional RNNs (BRNN) [25, 2] extend the unidirectional RNN by introducing a second hidden layer, where the hidden to hidden connections flow in opposite temporal order. The model is therefore able to exploit information both from the past and the future.

The output $\mathbf{y}_t$ is traditionally determined by

$$P\left(\mathbf{y}_t \mid \{\mathbf{x}_d\}_{d \neq t}\right) = \phi\left(\mathbf{W}_y^f \mathbf{h}_t^f + \mathbf{W}_y^b \mathbf{h}_t^b + \mathbf{b}_y\right),$$

but we propose the use of

$$P\left(\mathbf{y}_t \mid \{\mathbf{x}_d\}_{d \neq t}\right) = \phi\left(\mathbf{W}_y^f \mathbf{h}_{t-1}^f + \mathbf{W}_y^b \mathbf{h}_{t+1}^b + \mathbf{b}_y\right) \tag{3}$$

where

$$\mathbf{h}_t^f = \tanh\left(\mathbf{W}_h^f \mathbf{h}_{t-1}^f + \mathbf{W}_x^f \mathbf{x}_t + \mathbf{b}_h^f\right) \tag{4}$$

$$\mathbf{h}_t^b = \tanh\left(\mathbf{W}_h^b \mathbf{h}_{t+1}^b + \mathbf{W}_x^b \mathbf{x}_t + \mathbf{b}_h^b\right). \tag{5}$$

The structure of the BRNN is illustrated in Figure 1 (right). Compared with the regular RNN, the forward and backward directions have separate non-tied weights and hidden activations, and are denoted by the superscript f and b for forward and backward, respectively. Note that the connections are acyclic. Note also that in the proposed formulation, $\mathbf{y}_t$ does not get information from $\mathbf{x}_t$. We can therefore use the model in an unsupervised manner to predict one time step given all other time steps in the input sequence simply by setting $\mathbf{Y} = \mathbf{X}$.

# 3 Probabilistic Interpretation for Unsupervised Modelling

Probabilistic unsupervised modeling for sequences using a unidirectional RNN is straightforward, as the joint distribution for the whole sequence is simply the product of the individual predictions:

$$P_{\text{unidirectional}}(\mathbf{X}) = \prod_{t=1}^{T} P(\mathbf{x}_t \mid \{\mathbf{x}_d\}_{d=1}^{t-1}). \tag{6}$$

For the BRNN, the situation is more complicated. The network gives predictions for individual outputs given all the others, and the joint distribution cannot be written as their product. We propose two solutions for this, denoted by *GSN* and *NADE*.

**GSN** Generative Stochastic Networks (GSN) [6] use a denoising auto-encoder to estimate the data distribution as the asymptotic distribution of the Markov chain that alternates between corruption and denoising. The resulting distribution is thus defined only implicitly, and cannot be written analytically. We can define a corruption function that masks $\mathbf{x}_t$ as missing, and a denoising function that reconstructs it from the others. It turns out that one feedforward pass of the BRNN does exactly that.

Our first probabilistic interpretation is thus that the joint distribution defined by a BRNN is the asymptotic distribution of a process that replaces one observation vector $\mathbf{x}_t$ at a time in $\mathbf{X}$ by sampling it from $P_{\text{BRNN}}(\mathbf{x}_t \mid \{\mathbf{x}_d\}_{d \neq t})$. In practice, we will start from a random initialization and use Gibbs sampling.

**NADE** The Neural Autoregressive Distribution Estimator (NADE) [27] defines a probabilistic model by reconstructing missing components of a vector one at a time in a random order, starting from a fully unobserved vector. Each reconstruction is given by an auto-encoder network that takes as input the observations so far and an auxiliary mask vector that indicates which values are missing.

We extend the same idea for time series. Firstly, we concatenate an auxiliary binary element to input vectors to indicate a missing input. The joint distribution of the time series is defined by first drawing a random permutation $o_d$ of time indices $1 \ldots T$ and then setting data points observed one by one in that order, starting from a fully missing sequence:

$$P_{\text{NADE}}(\mathbf{X} \mid o_d) = \prod_{d=1}^{T} P(\mathbf{x}_{o_d} \mid \{\mathbf{x}_{o_e}\}_{e=1}^{d-1}). \tag{7}$$

In practice, the BRNN will be trained with some inputs marked as missing, while all the outputs are observed. See Section 5.1 for more training details.

# 4 Filling in gaps with Recurrent Neural Networks

The task we aim to solve is to fill in gaps of multiple consecutive data points in high-dimensional binary time series data. The inference is not trivial for two reasons: firstly, we reconstruct multiple consecutive data points, which are likely to depend on each other, and secondly, we fill in data in the middle of a time series and hence need to consider the data both before and after the gap.

For filling in gaps with the GSN approach, we first train a bidirectional RNN to estimate $P_{\text{BRNN}}(\mathbf{x}_t \mid \{\mathbf{x}_d\}_{d \neq t})$. In order to achieve that, we use the structure presented in Section 2.2. At test time, the gap is first initialized to random values, after which the missing values are sampled from the distribution $P_{\text{BRNN}}(\mathbf{x}_t \mid \{\mathbf{x}_d\}_{d \neq t})$ one by one in a random order repeatedly to approximate the stationary distribution. For the RNN structures used in this paper, the computational complexity of this approach at test time is $\mathcal{O}((dc + c^2)(T + gM))$ where $d$ is the dimensionality of a data point, $c$ is the number of hidden units in the RNN, $T$ is the number of time steps in the data, $g$ is the length of the gap and $M$ is the number of Markov chain Monte Carlo (MCMC) steps used for inference.

For filling in gaps with the NADE approach, we first train a bidirectional RNN where some of the inputs are set to a separate missing value token. At test time, all data points in the gap are first initialized with this token, after which each missing data point is reconstructed once until the whole gap is filled. Computationally, the main difference to GSN is that we do not have to sample each reconstructed data point multiple times, but the reconstruction is done in as many steps as there are missing data points in the gap. For the RNN structures used in this paper, the computational complexity of this approach at test time is $\mathcal{O}((dc + c^2)(T + g))$ where $d$ is the dimensionality of a data point, $c$ is the number of hidden units in the RNN, $g$ is the length of the gap and $T$ is the number of time steps in the data.

In addition to the two proposed methods, one can use a unidirectional RNN to solve the same task. We call this method *Bayesian MCMC*. Using a unidirectional RNN for the task of filling in gaps is not trivial, as we need to take into account the probabilities of the values after the gap, which the model does not explicitly do. We therefore resort to a similar approach as the GSN approach, where we replace the $P_{\text{BRNN}}(\mathbf{x}_t \mid \{\mathbf{x}_d\}_{d \neq t})$ with a unidirectional equivalent for the Gibbs sampling. As the unidirectional RNN models conditional probabilities of the form $P_{\text{RNN}}(\mathbf{x}_t \mid \{\mathbf{x}_d\}_{d=1}^{t-1})$, we can use Bayes' theorem to derive:

$$P_{\text{RNN}}\left(\mathbf{x}_t = \mathbf{a} \mid \{\mathbf{x}_d\}_{d \neq t}\right) \tag{8}$$

$$\propto P_{\text{RNN}}\left(\mathbf{x}_t = \mathbf{a} \mid \{\mathbf{x}_d\}_{d=1}^{t-1}\right) P_{\text{RNN}}\left(\{\mathbf{x}_e\}_{e=t+1}^{T} \mid \mathbf{x}_t = \mathbf{a}, \{\mathbf{x}_d\}_{d=1}^{t-1}\right) \tag{9}$$

$$= \prod_{\tau=t}^{T} P_{\text{RNN}}(\mathbf{x}_\tau \mid \{\mathbf{x}_d\}_{d=1}^{\tau-1})\Big|_{\mathbf{x}_t = \mathbf{a}} \tag{10}$$

where $P_{\text{RNN}}(\mathbf{x}_\tau \mid \{\mathbf{x}_d\}_{d=1}^{\tau-1})$ is directly the output of the unidirectional RNN given an input sequence $\mathbf{X}$, where one time step $t$, i.e. the one we Gibbs sample, is replaced by a proposal $\mathbf{a}$. The problem is that we have to go through all possible proposals $\mathbf{a}$ separately to evaluate the probability $P\left(\mathbf{x}_t = \mathbf{a} \mid \{\mathbf{x}_d\}_{d \neq t}\right)$. We therefore have to evaluate the product of the outputs of the unidirectional RNN for time steps $t \dots T$ for each possible $\mathbf{a}$.

In some cases this is feasible to evaluate. For categorical data, e.g. text, there are as many possible values for $\mathbf{a}$ as there are dimensions[1]. However, for other binary data the number of possibilities grows exponentially, and is clearly not feasible to evaluate. For the RNN structures used in this paper, the computational complexity of this approach at test time is $\mathcal{O}((dc + c^2)(T + aTM))$ where $a$ is the number of different values a data point can have, $d$ is the dimensionality of a data point, $c$ is the number of hidden units in the RNN, $T$ is the number of time steps in the data, and $M$ is the number of MCMC steps used for inference. The critical difference in complexity to the GSN approach is the coefficient $a$, that for categorical data takes the value $d$, for binary vectors $2^d$ and for continuous data is infinite.

As a simple baseline model, we also evaluate the *one-gram* log-likelihood of the gaps. The one-gram model assumes a constant context-independent categorical distribution for the categorical task, or a

vector of factorial binomial probabilities for the structured prediction task:

$$P_{\text{one-gram}}\left(\mathbf{y}_t\right) = \text{f}\left(\mathbf{b}_y\right).$$

This can be done in $\mathcal{O}(dg)$.

We also compare to *one-way inference*, where the data points in the gap are reconstructed in order without taking the future context into account, using Equations (1) and (2) directly. The computational complexity is $\mathcal{O}((dc + c^2)T)$.

## 5 Experiments

We run two sets of experiments: one for a categorical prediction task, and one for a binary structured prediction task. In the categorical prediction task we fill in gaps of five characters in Wikipedia text, while in the structural prediction task we fill in gaps of five time steps in different polyphonic music data sets.

### 5.1 Training details for categorical prediction task

For the categorical prediction task, we test the performance of the two proposed methods, GSN and NADE. In addition, we compare the performance to MCMC using Bayesian inference and one-way inference with a unidirectional RNN. We therefore have to train three different RNNs, one for each method.

Each RNN is trained as a predictor network, where the character at each step is predicted based on all the previous characters (in the case of the RNN) or all the previous and following characters (in the case of the BRNNs). We use the same data set as Sutskever *et al.* [26], which consists of 2GB of English text from Wikipedia. For training, we follow a similar strategy as Hermans and Schrauwen [15]. The characters are encoded as one-hot binary vectors with a dimensionality of $d = 96$ characters and the output is modelled with a softmax distribution. We train the unirectional RNN with string lengths of $T = 250$ characters, where the error is propagated only from the last 200 outputs. In the BRNN we use string length of $T = 300$ characters, where the error is propagated from the middle 200 outputs. We therefore avoid propagating the gradient from predictions that lack long temporal context.

For the BRNN used in the NADE method, we add one dimension to the one-hot input which corresponds to a missing value token. During training, in each minibatch we mark $g = 5$ consecutive characters every 25 time steps as a gap. During training, the error is propagated only from these gaps. For each gap, we uniformly draw a value from 1 to 5, and set that many characters in the gap to the missing value token. The model is therefore trained to predict the output in different stages of inference, where a number of the inputs are still marked as missing. For comparison, we also train a similar network, but without masking. In that variant, the error is therefore propagated from all time steps. We refer to "NADE" masked and "NADE no mask", respectively, for these two training methods.

For all the models, the weight elements are drawn from the uniform distribution: $w_{i,j} \sim \mathcal{U}\left[-s, s\right]$ where $s = 1$ for the input to hidden layer, and following Glorot and Bengio [10], where $s = \sqrt{6/\left(d_{in} + d_{out}\right)}$ for the hidden-to-hidden and the hidden-to output layers. The biases are initialized to zero.

We use $c = 1000$ hidden units in the unidirectional RNN and $c = 684$ hidden units in the two hidden layers in the BRNNs. The number of parameters in the two model types is therefore roughly the same. In the recurrent layers, we set the recurrent activation connected to the first time step to zero.

The networks are trained using stochastic gradient descent and the gradient is calculated using backpropagation through time. We use a minibatch size of 40, i.e. each minibatch consists of 40 randomly sampled sequences of length 250. As the gradients tend to occasionally "blow up" when training RNNs [5, 20], we normalize the gradients at each update to have length one. The step size is set to 0.25 for all layers in the beginning of training, and it is linearly decayed to zero during training. As training the model is very time-consuming[2], we do not optimize the hyperparameters, or repeat runs to get confidence intervals around the evaluated performances.

## 5.2 Training Details for the Binary Structured Prediction Task

In the other set of experiments, we use four polyphonic music data sets [8]. The data sets consist of at least 7 hours of polyphonic music each, where each data point is a binary $d = 88$-dimensional vector that represents one time step of MIDI-encoded music, indicating which of the 88 keys of a piano are pressed. We test the performance of the two proposed methods, but omit training the unidirectional RNNs as the computational complexity of the Bayesian MCMC is prohibitive ($a = 2^{88}$).

We train all models for 50 000 updates in minibatches of $\approx 3\,000$ individual data points[3]. As the data sets are small, we select the initial learning rate on a grid of $\{0.0001, 0.0003, \ldots, 0.3, 1\}$ based on the lowest validation set cost. We use no "burn-in" as several of the scores are fairly short, and therefore do not specifically mask out values in the beginning or end of the data set as we did for the text data.

For the NADE method, we use an additional dimension as a missing value token in the data. For the missing values, we set the missing value token to one and the other dimensions to zero.

Other training details are similar to the categorical prediction task.

## 5.3 Evaluation of Models

At test time, we evaluate the models by calculating the mean log-likelihood of the correct value of gaps of five consecutive missing values in test data.

In the GSN and Bayesian MCMC approaches, we first set the five values in the gap to a random value for the categorical prediction task, or to zero for the structured prediction task. We then sample all five values in the gap in random order, and repeat the procedure for $M = 100$ MCMC steps[4]. For evaluating the log-likelihood of the correct value for the string, we force the last five steps to sample the correct value, and store the probability of the model sampling those values. We also evaluate the probability of reconstructing correctly the individual data points by not forcing the last five time steps to sample the correct value, but by storing the probability of reconstructing the correct value for each data point separately. We run the MCMC chain 100 times and use the log of the mean of the likelihoods of predicting the correct value over these 100 runs.

When evaluating the performance of one-directional inference, we use a similar approach to MCMC. However, when evaluating the log-likelihood of the entire gap, we only construct it once in sequential order, and record the probabilities of reconstructing the correct value. When evaluating the probability of reconstructing the correct value for each data point separately, we use the same approach as for MCMC and sample the gap 100 times, recording for each step the probability of sampling the correct value. The result for each data point is the log of the mean of the likelihoods over these 100 runs.

On the Wikipedia data, we evaluate the GSN and NADE methods on 50 000 gaps on the test data. On the music data, all models are evaluated on all possible gaps of $g = 5$ on the test data, excluding gaps that intersect with the first and last 10 time steps of a score. When evaluating the Bayesian MCMC with the unidirectional RNN, we have to significantly limit the size of the data set, as the method is highly computationally complex. We therefore run it on 1 000 gaps on the test data.

For NADE, we set the five time steps in the gap to the missing value token. We then reconstruct them one by one to the correct value, and record the probability of the correct reconstruction. We repeat this process for all possible permutations of the order in which to do the reconstruction, and therefore acquire the exact probability of the correct reconstruction given the model and the data. We also evaluate the individual character reconstruction probabilities by recording the probability of sampling the correct value given all other values in the gap are set to missing.

## 5.4 Results

From Table 1 we can see that the Bayesian MCMC method seems to yield the best results, while GSN or NADE outperform one-way inference. It is worth noting that in the most difficult data sets,

Table 1: Negative Log Likelihood (NLL) for gaps of five time steps using different models (lower is better). In the experiments, GSN and NADE perform well, although they are outperformed by Bayesian MCMC.

| Inference strategy | Wikipedia | Nottingham | Piano | Muse | JSB |
|---|---|---|---|---|---|
| GSN | 4.60 | 19.1 | **38.8** | 37.3 | **43.8** |
| NADE masked | 4.86 | 19.0 | 40.4 | **36.5** | 44.3 |
| NADE | 4.88 | **18.5** | 39.4 | 34.7 | 44.6 |
| Bayesian MCMC | **4.37** | NA | NA | NA | NA |
| One-way inference | 5.79 | 19.2 | 38.9 | 37.6 | 43.9 |
| One-gram | 23.3 | 145 | 138 | 147 | 118 |

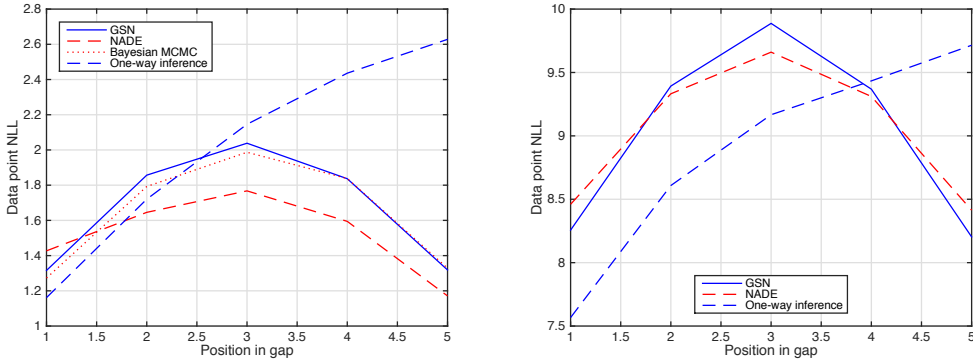

Figure 2: Average NLL per data point using different methods with the Wikipedia data set (left) and the Piano data set (right) for different positions in a gap of 5 consecutive missing values. The middle data point is the most difficult to estimate for the most methods, while the one-way inference cannot take future context into account making prediction of later positions difficult. For the left-most position in the gap, the one-way inference performs the best since it does not require any approximations such as MCMC.

piano and JSB, oneway inference performs very well. Qualitative examples of the reconstructions obtained with the GSN and NADE on the Wikipedia data are shown in Table 3 (supplementary material).

In order to get an indication of how the number of MCMC steps in the GSN approach affects performance, we plotted the difference in NLL of GSN and NADE of the test set as a function of the number of MCMC steps in Figure 3 (supplementary material). The figure indicates that the music data sets mix fairly well, as the performance of GSN quickly saturates. However, for the Wikipedia data, the performance could probably be even further improved by letting the MCMC chain run for more than $M = 100$ steps.

In Figure 2 we have evaluated the NLL for the individual characters in the gaps of length five. As expected, all methods except for one-way inference are better at predicting characters close to both edges of the gap.

As a sanity check, we make sure our models have been successfully trained by evaluating the mean test log-likelihood of the BRNNs for gap sizes of one. In Table 2 (supplementary material) we can see that the BRNNs expectedly outperform previously published results with unidirectional RNNs, which indicates that the models have been trained successfully.

## 6   Conclusion and Discussion

Although recurrent neural networks have been used as generative models for time series data, it has not been trivial how to use them for inference in cases such as missing gaps in the sequential data.

In this paper, we proposed to use bidirectional RNNs as generative models for time series, with two probabilistic interpretations called GSN and NADE. Both provide efficient inference in both positive and negative directions in time, and both can be used in tasks where Bayesian inference of a unidirectional RNN is computationally infeasible.

The model we trained for NADE differed from the basic BRNN in several ways: Firstly, we artificially marked gaps of 5 consecutive points as missing, which should help in specializing the model for such reconstruction tasks. It would be interesting to study the effect of the missingness pattern used in training, on the learned representations and predictions. Secondly, in addition to using all outputs as the training signal, we tested using only the reconstructions of those missing values as the training signal. This reduces the effective amount of training that the model went through. Thirdly, the model had one more input (the missingness indicator) that makes the learning task more difficult. We can see from Table 2 that the model we trained for NADE where we only used the reconstructions as the training signal has a worse performance than the BRNN for reconstructing single values. This indicates that these differences in training have a significant impact on the quality of the final trained probabilistic model.

We used the same number of parameters when training an RNN and a BRNN. The RNN can concentrate all the learning effort on forward prediction, and re-use the learned dependencies in backward inference by the computationally heavy Bayesian inference. It remains an open question which approach would work best given an optimal size of the hidden layers.

As future work, other model structures could be explored in this context, for instance the Long Short-Term Memory [16]. Specifically to our NADE approach, it might make sense to replace the regular additive connection from the missingness indicator input to the hidden activations in Eq. (4,5), by a multiplicative connection that somehow gates the dynamics mappings $\mathbf{W}_h^f$ and $\mathbf{W}_h^b$. Another direction to extend is to use a deep architecture with more hidden layers.

The midi music data is an example of a structured prediction task: Components of the output vector depend strongly on each other. However, our model assumes independent Bernoulli distributions for them. One way to take those dependencies into account is to use stochastic hidden units $\mathbf{h}_t^f$ and $\mathbf{h}_t^b$, which has been shown to improve performance on structured prediction tasks [22]. Bayer and Osendorfer [4] explored that approach, and reconstructed missing values in the middle of motion capture data. In their reconstruction method, the hidden stochastic variables are selected based on an auxiliary inference model, after which the missing values are reconstructed conditioned on the hidden stochastic variable values. Both steps are done with maximum a posteriori point selection instead of sampling. Further quantitative evaluation of the method would be an interesting point of comparison.

The proposed methods could be easily extended to continuous-valued data. As an example application, time-series reconstructions with a recurrent model has been shown to be effective in speech recognition especially under impulsive noise [23].

## Acknowledgements

We thank KyungHyun Cho and Yoshua Bengio for useful discussions. The software for the simulations for this paper was based on Theano [3, 7]. Nokia has supported Mathias Berglund and the Academy of Finland has supported Tapani Raiko.

## Footnotes

[1]For character-based text, the number of dimensions is the number of characters in the model alphabet.

[2]We used about 8 weeks of GPU time for the reported results.

[3]A minibatch can therefore consist of e.g. 100 musical scores, each of length $T = 30$.

[4]$M = 100$ MCMC steps means that each value in the gap of $g = 5$ will be resampled $M/g = 20$ times

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
