[Supplementary Material]



Figure 3: Difference in gap negative test log-likelihood between GSN and NADE for different data sets. We can see that GSN outperforms NADE after a certain threshold of MCMC steps. Note that the rising curves with music data sets might be explained by the model being overfitted to the training set.

Table 2: Negative Log Likelihood (NLL) over the test data for the trained models. As a comparison, similar numbers are presented for the unidirectional RNNs trained by Boulanger-Lewandowski *et al.* [8] and by Hermans and Schrauwen [15]. The results serve as a sanity check that the trained models have been trained successfully. Note that the BRNN has more context than the unidirectional RNNs, and is hence expected to perform better measured by NLL. Also note that the training of the model for NADE without masking is very similar to the training of the BRNN.

| Inference strategy | Wikipedia | Nottingham | Piano | Muse | JSB |
|---|---|---|---|---|---|
| NLL of BRNN | **0.37** | 3.23 | **6.82** | 5.68 | 7.80 |
| NLL of NADE masked | 0.55 | 3.32 | 7.42 | 6.48 | 8.51 |
| NLL of NADE | 0.41 | **2.89** | 7.05 | **5.54** | **7.59** |
| NLL of unidirectional RNN | 1.21 | 3.87 | 7.88 | 7.43 | 8.76 |
| Boulanger-Lewandowski *et al.* [8] | | 4.46 | 8.37 | 8.13 | 8.71 |
| Hermans and Schrauwen [15] | 1.12 | | | | |

Table 3: Random samples from reconstructed gaps (underlined) using either NADE (left) or GSN (right). Note that double spaces are difficult to distinguish from single spaces.

| NADE | GSN |
|---|---|
| s practice their short for as long as possibl | s practice their show,d ar as long as possibl |
| nd nephews through copiene. It was reported o | nd nephews through clubs.e. It was reported o |
| but originally a nuclear bunker to protect th | but originally a nuclear bunker to protect th |
| the Opera" and "The Heas", have been fully r | the Opera" and "Thereove", have been fully r |
| e III. Dunmore died in March 1809 and was suc | e III. Dunmore died in March 1809 and was suc |
| ch fades from golden aillow at the center thr | ch fades from goldenly show at the center thr |
| Colorado state champion who is credited with | Colorado state champions ho is credited with |
| ing Bushroot, Liquida. HL and Megavolt). His | ing Bushroot, Liquidlands and Megavolt). His |
| acial lake bed known as the Black Dirt Region | acial lake bed known of the Black Dirt Region |
| e all ancient leadersidred it to laud their a | e all ancient leader cannd it to laud their a |
| ted November 2005. They series also featured | ted November 2005. The series also featured |
| TR amyloid is extractedliar. Treatment of TT | TR amyloid is extract s war. Treatment of TT |
| hile the gigantic "Ston saurus sikanniensis", | hile the gigantic "S"So Saurus sikanniensis", |
| area. Initially one other compartment was an | area. Initially one other compartment was an |