[Reviews · NeurIPS 2015]

Submitted by Assigned_Reviewer_1

The authors present a way of using bidirectional RNNs for 'gap-filling' tasks, i.e. for smoothing, where one needs to consider the data both before and after the gap. For this they provide two solutions, one where the bidirectional RNN is interpreted in a way that is inspired by generative stochastic networks (GSN), and a second where it is interpreted in a way that is similar to the neural autoregressive distribution estimators (NADE). In experiments they show that both interpretations lead to gap-fillings that are more accurate in terms of NLL than those generated by a unidirectional RNN, but less accurate than Bayesian inference. However since the method is much less expensive than Bayesian inference, it enables them to work with more complex datasets.

The paper is well written, the ideas are novel to my knowledge and the results are strong.

I think it will be beneficial for the paper for it to have a diagram visualising the notation, specifically where the 'gap' is, and what g, a, c, d, T, M and so on are. In its current form the notation is slightly difficult to remember.

Line 170: in a row -> consecutively Line 267: "blow up" -> ``blow up" Line 284: "burn in" -> ``burn in" Line 409: midi -> MIDI
Summary: The authors provide two novel interpretations of bidirectional RNNs that allow them to be used as generative models. The experimental evaluation is thorough, the results are strong and the paper is well presented. This work is likely to be of high interest for the machine learning community.

Submitted by Assigned_Reviewer_2

Summary:

The paper proposes a modification of the traditional bidirectional RNN, which predicts output at time t from forward hidden state t-1 and backward hidden state t+1, without getting information from x_t. This allows training the network to map directly from inputs X to outputs Y. The authors test the capability of this architecture to compute the likelihood of completed gaps in sequences, using GSN and NADE - the results are compared to "Bayesian MCMC" inference in standard RNNs. The paper shows that for longer gaps, GSN using the specified BRNN architecture is computationally cheaper and competitive in quality with RNNs and NADE is inferior in quality to GSN.

Quality and Clarity The models and the inference, which was used to compute the NLL scores, are both clearly described. I did not see major technical or experimental setup issues.

- The "one-way inference" baseline seems to outperform the BRNN for the first couple of characters (Fig 2, left side). Why is it missing in the right plot?

- Given the good performance of "one-way inference" on the first couple of characters (Fig 2), wouldn't it make sense to also compare to the setup where we have 2 RNNs, one trained to predict characters going forward, one trained to predict going backward, and then we pick the first two characters from the forward RNN, last two characters from the backward RNN and perhaps the middle by voting? How would this baseline have done? This naive approach may well break down for longer sequences, but if it dominates it would suggest the experimental setup being "too easy" -- a gap of 5 characters may not be sufficient to show significant BRNN benefits.

- Why is GSN better than NADE on wikipedia in Fig 1 yet appearing noticeably worse for wikipedia in Fig 2?

- For Bayesian MCMC on RNNs (eq 10), when the gap is large making it hard to explore every single variable setting, is it possible to approximately estimate the posterior using direct MCMC sampling in the RNN rather than explicitly enumerating all possible missing states?

Originality The modification of BRNN that connects each output y_t to forward hidden state x_{t-1} and to backward hidden state x_{t+1} appears novel to the best of my knowledge. It is a rather straightforward extension of the standard RNN. I am also not aware of existing experiments on evaluating the performance of such a model for likelihood estimation of sequence gaps task, although the integration of the BRNN with GSN and NADE is rather straightforward.

Significance I am not convinced that the task of estimating the likelihood of sequence completions has much practical interest. While it is a convenient way to evaluate model quality, getting MAP estimates (e.g. using beam search) may be sufficient for most practical uses?

While the BRNN is faster than the Bayesian MCMC, the quality seems a bit inferior. I am curious if there is a way to achieve getting the estimates using MCMC and RNNs without enumerating all states? That would be a reasonable alternative.

Summary: The paper proposes a BRNN architecture variant and demonstrates that it can be integrated into GSN or NADE to estimate the likelihood of filled-in gaps in sequences. The approach appears novel, while a bit straightforward. I do not think the task addressed, at least in the concrete experimental incarnation, is particularly relevant in practice. The main gain of the proposed approach is the ability to get reasonable likelihood estimates for the completions in an efficient way, but the chosen baselines for the task seem perhaps a bit too easy.

Submitted by Assigned_Reviewer_3

This paper investigates a number of methods for generating in bidirectional settings, such as a language model where a left and a right hand side context is given. For this it makes use of two algorithms, GSN as proposed by Bengio et al. and NADE as developed by Larochelle and Murray.

The paper sets out by recapping these two algorithms and then proposes a mechanism for using GSN and NADE for gap-filling in time sequence models. As already stated in my summary above, I don't think there is huge novelty or innovation in this approach, but the strength of this paper lies in the comprehensive explanation of both algorithms, as well as the subsequent experimental section where both methods as well as a number of plausible baselines are evaluated on two time-series tasks, which I found very informative.

My main criticism is that the authors could demarcate their own contribution from the prior work more clearly, by making explicit what is novel and what simply a reformulation of prior work. For instance, Larochelle and Murray already discuss generative properties of NADE in the original paper, which isn't clear from reading this paper.

On the experimental front, I understand the benefit of only using the error propagated from missing values in the NADE setting, but would like to also see results in a setting where a signal is backpropagated from all values in the auto-encoded time sequence.
Summary: What the paper may lack in novelty by effectively applying existing algorithms to a new problem, it makes up for in rigorous analysis and discussion. I learned something from reading this paper and think it is worthy of publication.

Author Feedback
Author rebuttal: Thank you for the valuable reviews. Please see the comments below to specific points raised in the reviews.

Reviewer 1 & 3

Thank you for the comments, we will revise the paper in accordance with the proposed clarifications.

"On the experimental front, I understand the benefit of only using the error propagated from missing values in the NADE setting, but would like to also see results in a setting where a signal is backpropagated from all values in the auto-encoded time sequence."
We also tried this approach, but left it out as it performed worse than the current implementation. We will comment on this in the revised version of the paper.

Reviewer 2

"Why is it [one-way inference] missing in the right plot? "
Thank you for the comment, we will add it to the plot on the Piano dataset.

"Given the good performance of "one-way inference" on the first couple of characters (Fig 2), wouldn't it make sense to also compare to the setup where we have 2 RNNs, one trained to predict characters going forward, one trained to predict going backward, and then we pick the first two characters from the forward RNN, last two characters from the backward RNN and perhaps the middle by voting?"
Thank you for the suggestion. It is worth noting that the bidirectional model has roughly the same number of parameters as the unidirectional RNN, which means that there are less parameters "per direction".

"Why is GSN better than NADE on wikipedia in Fig 1 yet appearing noticeably worse for wikipedia in Fig 2?"
(We assume that the reviewer refers to Table 1 vs. Fig 2). The number in Table 1 refers to the log likelihood of the entire gap, while Figure 2 depicts the datapoint-wise likelihoods. Therefore these two measures are not entirely comparable. We will revise the text to make this point clearer.

"For Bayesian MCMC on RNNs (eq 10), when the gap is large making it hard to explore every single variable setting, is it possible to approximately estimate the posterior using direct MCMC sampling in the RNN rather than explicitly enumerating all possible missing states?"
For a unidirectional RNN, it is still not clear how to take into account the values of time steps larger than t when sampling data point at time t without enumerating all possible states for step t. Therefore, it is unfortunately not trivial to even sample from the correct distribution without enumerating all states.

"I am not convinced that the task of estimating the likelihood of sequence completions has much practical interest. While it is a convenient way to evaluate model quality, getting MAP estimates (e.g. using beam search) may be sufficient for most practical uses? "
It is indeed true that for some practical uses, it is sufficient to draw a few samples. However, e.g. beam search does not necessarily work well in this setup (see our comments for Reviewer 5). We will elaborate on this point clearer in the revised version of the paper.

Reviewer 5

"I am wondering why the authors did not consider a beam-search reconstruction as another baseline"
In our preliminary analyses we experimented with the beam search approach, but left it out as there are quite a few issues with the method when reconstructing gaps of more than one time step. Specifically, the number of beams had to be set to a fairly high value in order for the method to perform decently compared to the other methods even for a gap size of five time steps. In addition, it is not trivial to estimate the log likelihood of the method, and it would hence have to be compared using some other measure (e.g. accuracy). What we try to argue in the paper is that it makes more sense to train a bidirectional RNN in the first place instead of resorting to e.g. beam search. We will elaborate more on this point in the revised version of the paper.

"Overall my impression is that the paper can be much improved with more powerful models (read LSTMs, attention-based mechanisms, memory/stack networks) and also larger numerical experiments."
These structural additions to the RNNs would likely have improved the results of all methods. However, we deliberately chose not to use more complex RNN structures in order not to distract the reader from the main points of the paper.

Reviewer 6

"It is not too novel a contribution especially given the original idea is present in a paper from 1997."
The idea of a bidirectional RNN was indeed proposed already in 1997, but that work does not discuss reconstructing gaps of more than one time step. The novelty in our research specifically lies in studying methods for reconstructing gaps of size larger than one time step.